# Insulin-Like Growth Factor 2 (IGF2) Signaling in Colorectal Cancer—From Basic Research to Potential Clinical Applications

**DOI:** 10.3390/ijms20194915

**Published:** 2019-10-03

**Authors:** Aldona Kasprzak, Agnieszka Adamek

**Affiliations:** 1Department of Histology and Embryology, University of Medical Sciences, Swiecicki Street 6, 60-781 Poznan, Poland; 2Department of Infectious Diseases, Hepatology and Acquired Immunodeficiencies, University of Medical Sciences, Szwajcarska Street 3, 61-285 Poznan, Poland; agnieszkaadamek@ump.edu.pl

**Keywords:** colorectal cancer (CRC), insulin growth factor 2 (IGF2), genetic and epigenetic changes, IGF2-associated biomarkers in CRC

## Abstract

Colorectal cancer (CRC) is one of the most common cancers in men and women worldwide as well as is the leading cause of death in the western world. Almost a third of the patients has or will develop liver metastases. While genetic as well as epigenetic mechanisms are important in CRC pathogenesis, the basis of the most cases of cancer is unknown. High spatial and inter-patient variability of the molecular alterations qualifies this cancer in the group of highly heterogeneous tumors, which makes it harder to elucidate the mechanisms underlying CRC progression. Determination of highly sensitive and specific early diagnosis markers and understanding the cellular and molecular mechanism(s) of cancer progression are still a challenge of the current era in oncology of solid tumors. One of the accepted risk factors for CRC development is overexpression of insulin-like growth factor 2 (IGF2), a 7.5-kDa peptide produced by liver and many other tissues. IGF2 is the first gene discovered to be parentally imprinted. Loss of imprinting (LOI) or aberrant imprinting of *IGF2* could lead to IGF2 overexpression, increased cell proliferation, and CRC development. IGF2 as a mitogen is associated with increased risk of developing colorectal neoplasia. Higher serum IGF2 concentration as well as its tissue overexpression in CRC compared to control are associated with metastasis. IGF2 protein was one of the three candidates for a selective marker of CRC progression and staging. Recent research indicates dysregulation of different micro- and long non-coding RNAs (miRNAs and lncRNAs, respectively) embedded within the *IGF2* gene in CRC carcinogenesis, with some of them indicated as potential diagnostic and prognostic CRC biomarkers. This review systematises the knowledge on the role of genetic and epigenetic instabilities of *IGF2* gene, free (active form of IGF2) and IGF-binding protein (IGFBP) bound (inactive form), paracrine/autocrine secretion of IGF2, as well as mechanisms of inducing dysplasia in vitro and tumorigenicity in vivo. We have tried to answer which molecular changes of the *IGF2* gene and its regulatory mechanisms have the most significance in initiation, progression (including liver metastasis), prognosis, and potential anti-IGF2 therapy in CRC patients.

## 1. Introduction

Globally, colorectal cancer (CRC) is one of the most common and severe malignant tumors in humans. According to the most recent data, CRC is third tumor worldwide in terms of incidence but second in terms of mortality [1,2]. The global burden of CRC is expected to increase by 60% to more than 2.2 million new cases and 1.1 million deaths by 2030 [3]. In particular, high relapse rates and hepatic metastases are the major contributors of CRC patient mortality [4,5]. 

While the main signaling pathways and stages of CRC progression have already been described, high inter-patient variability and high spatial heterogeneity of the tumor is a constant obstacle in design of new effective forms of therapy [6,7,8,9,10]. Around 90% of CRC cases develop sporadically, with only a few (less than 10%) attributed to hereditary origin [11]. While the pathogenetic mechnisms of CRC in inflammatory bowel disease (IBD) are poorly described, IBD is also considered as a factor of CRC development [12]. 

The Cancer Genome Atlas (TCGA) also contains an examination of CRC tumors [6,7,13,14,15]. In this way (2013), three molecular types of CRC were identified: hypermutated (13%), ultra-mutated (3%), and with chromosomal instability (CIN) (84%). More than 80% of sporadic CRC is characterized as CIN, characterized by changes in chromosome structure and number that include gains, deletions, translocations, and other chromosomal rearrangements. The next group in terms of incidence (~13–16% of sporadic CRC) is characterized by hypermutations and shows microsatellite instability (MSI) caused by faulty DNA mismatch repair (MMR), which is usually connected to wild-type *TP53* and chromosomal instability of a near-diploid pattern (reviewed in Reference [7]). According to the consensus molecular subtypes (CMS) (2015), four groups are included in the classification based on expression signatures: CMS1 (MSI-immune, 14%), CMS2 (canonical, 37%), CMS3 (metabolic, 13%), and CMS4 (mesenchymal, 23%), with the residual unclassified group (mixed features, 13%) containing the remaining cases [13]. Almost all hypermutated MSI cancers fall into the first category (CMS1), with the remaining microsatellite stable (MSS) cancers subcategorized into the remaining three groups [7]. Some studies distinguish a poor-prognosis stem/serrated/mesenchymal (SSM) transcriptional subtype of CRC, characterized by abundant stromal component [16], as well as five new CRC intrinsic subtypes (CRIS) endowed with distinctive molecular, functional, and phenotypic features [17]. 

Colorectal adenoma-carcinoma sequence observed in most of the CRC cases in humans (CIN tumors, 84%) is associated with high frequency of DNA somatic copy number alterations (SCNA), with common mutations in *APC*, *TP53*, *KRAS*, *SMAD4*, and *PIK3CA* [7,10,18,19,20]. On the other hand, serrated neoplastic pathways associated with *BRAF* mutations, *hMLH1* promoter hypermethylation, as well as MSI constitute 14–30% of CRC cases [21,22].

While genetic [21,22,23,24] and epigenetic [11,25,26,27,28,29] mechanisms are indisputable in colorectal carcinogenesis, the basis and regulatory mechanisms of the most cases of cancer are unknown [30]. 

According to traditional, genetic, and linear model of colorectal tumorigenesis, CRC develops as a result of mutational activation of oncogenes coupled with the inactivation of tumor suppressor genes [18,19,31,32]. Advances in gene and protein sequencing technology, bioinformatics, and/or biostatistical analyses allow not only for verification of different classification methods (CMS and CRIS) but also for expansion of the list of diagnostic-prognostic biomarkers, as well as development of more effective CRC therapies [7,16,17,27,29,33,34]. 

The last decade is dominated by studies indicating dysregulation of different long non-coding and microRNAs (lncRNAs and miRNAs, respectively) in colorectal carcinogenesis, with some of them indicated as potential diagnostic and prognostic CRC biomarkers [30,33,35,36,37]. CRC is one of the cancer types responsible for remarkable achievements in lncRNA research [38,39,40,41]. It was proven that some of these forms of RNA (e.g., lncRNAs 91H, PVT-1, and MEG3) can serve as biomarkers of improved sensitivity in early-stage CRC compared to the combination of CEA and CA19-9—the biomarkers currently used for CRC detection [42]. Three groups of miRNAs, oncogenic, tumor suppressive, and regulatory, were also implicated in CRC [43]. Remarkably, colon-adenocarcinoma-specific mRNAs, miRNAs, and lncRNAs were also identified [15]. The clinical significance (diagnostic-prognostic response to therapy) of numerous miRNAs in CRCs is also confirmed by meta-analyses [44,45]. The recent data of Dai et al. (2019) suggests that 4307 lncRNAs and 5798 mRNAs are deregulated during CRC initiation [46].

For many decades, a relation is noted between different components of the insulin-like growth factor (IGF) system and the development of cancers, both solid tumors and hematological malignancies [47,48,49,50]. As polypeptide growth factors (GFs), IGFs are some of the key regulators of different stages of cancer progression, being implicated in initiating carcinogenesis as well as tumor development and metastasis [47,48,51]. Elevated expression of IGF1 and IGF2 in CRC as well as co-existence of many metabolic disorders in patients affected by this malignancy (e.g., adiposity, dyslipidemia, hyperinsulinemia, and altered glucose homeostasis) allows to classify the elements of the IGF system as biomarkers of CRC predisposition and prognosis in human [52,53]. Obesity increases blood insulin levels and decreases IGFBP1 concentration, resulting in a rise in IGF1 levels [54]. Lack of physical activity (PA) has been specifically linked to CRC mortality, with ~15% of CRC deaths worldwide. Decreasing levels of IGF1 and increasing levels of IGFBP3 are presumable mechanisms underlying the inverse association between PA and CRC survival [55]. 

Using a qPCR-based technique, *IGF2* was included to the five genes with the highest average hypermethylated percentages (50.4%) in the CRC group. All these genes were observed to have the greatest potential of gene expression repression [56]. Genetic factors are also responsible for higher interindividual variation of the circulating levels of IGF2 (66%) compared to IGF1 (38%) [57,58]. A modulating influence of genetic variants of *IGF2 receptor* (*IGF2R)* (*IGF2R* c.5002 G > A and *IGF2R* c.901 C > G variants) on higher circulating IGF2 levels was also proven [59]. Based on TCGA data, it was proven that, among mutated genes typical for CRC (e.g., *APC, TP53, SMAD4, PIK3CA*, and *KRAS*) [5,19,20], there were also copy-number changes including amplifications of *ERBB2* and *IGF2* [6,29]. Recent data also describes the CRIS-D subtype, which includes Wingless/Integrated (WNT) pathway activation as well as *IGF2* overexpression and amplification. The analysis of ~300 publicly available CRC data sets revealed a trend for better prognosis (longer disease-free survival) in cases of CRIS-D tumors [17]. 

Considering the role of *IGF2* in colon carcinogenesis, genetic and epigenetic factors are mainly responsible for IGF2 overexpression [10,11,24,29,58,60,61,62,63]. *IGF2* transcription is regulated by genomic imprinting, achieved through methylation of the differentially methylated region (DMR) on the maternal allele [64,65,66], which contains expression of this gene in most tissues to the paternal allele. Loss of imprinting (LOI) of *IGF2* could lead to IGF2 overexpression, increased cell proliferation, and tumor development [11,28,56,62,67,68,69]. In a small fraction of patients (13%) heterozygous for *IGF2* DNA in nontumorous tissues, loss of heterozygosity (LOH) was also described [70]. The molecular mechanisms of high-level *IGF2* gene activation in CRC are constantly researched [24]. In turn, the role of *IGF2* genomic aberrations in the metastatic cascade model of CRC, still immensely difficult to elucidate, remains an open matter [10].

The participation of environmental factors combined with the action of the IGF2 signaling pathway in CRC development is much less definitive. The roles of obesity, improper diet, as well as exposition to certain drugs are considered in the development of CRC pathogenesis. However, the particular mechanisms of IGF2 connection to these factors are unclear. Among others, it was proven that consumption of ≥15 g alcohol/day was associated with exclusively elevated risk of CRC with *IGF2* DMR0 hypomethylation but not risk of cancer with high-level *IGF2* DMR0 methylation, suggesting a possible mechanistic link between alcohol intake and CRC risk through *IGF2* DMR0 hypomethylation during colorectal carcinogenesis [71]. The connections between IGF2 and type II diabetes mellitus (T2DM) and non-islet cell tumor hypoglycemia (NICTH) in etiology of this cancer are also examined [52,72,73]. Common IGF2 expression by tumor cells as well as autocrine action of this growth factor were also brought to attention, occasionally even reaching target tissues and causing tumor-induced hypoglycemia [74]. While some studies did not show quantitative differences in IGF2 expression in CRC patients with T2DM, compared to those not affected by diabetes [75], newer research report that upregulation of DOK5, IGF2, and IRS2 in colorectal cancer and T2DM patients is associated with significantly decreased overall survival (OS) [73]. A role of IGF2 is implicated in preadipocyte differentiation and metabolism. The results indicate a protective action of IGF2 on visceral fat deposition in children [76].

The role of IGF2 in CRC etiology and pathogenesis is indicated by epidemiological data and studies on tissue IGF2 expression in colorectal tumors. People with serum IGF2 levels in the upper quartile of the normal range (and IGFBP3 in the lower quartile) have a higher relative risk for developing cancer, including colon cancer [74]. Higher serum IGF2 concentrations [77,78], as well as positive correlation of these concentrations with more advanced CRC is noted in CRC patients compared to control [79,80]. Correlation between high levels of IGF1 and IGF2 and significantly increased CRC risk is also confirmed by meta-analysis [81]. Tissue IGF2 overexpression is also observed in primary CRC [82,83,84,85,86,87] as well as in in vitro models, suggesting autocrine action of this polypeptide [88,89,90,91]. In turn, in CRC metastasis tumors (mainly liver metastases), the levels of IGF2 expression in more varied, being found to be both overexpressed [92] and under-expressed compared with either colorectal primary tumors or adjacent normal mucosae [87]. IGF2 became one of the three candidates (with TGF-α and MMP-2) for selective markers of progression, a marker of a tumor staging [93,94], as well as a key factor in the early stage of colorectal carcinogenesis [60].

Among differentially expressed genes (DEGs), the value of lncRNA *IGF2* antisense (*IGF2-AS*) expression was confirmed, as a helpful CRC diagnostic-prognostic factor [15,34].

This study aims to evaluate the role of IGF2 in colorectal carcinogenesis, as its etiological connection to many cancers was indicated and no review focused solely on colorectal cancer can currently be found. We have tried to answer which molecular changes of the *IGF2* gene have the most significance in initiation, progression (including liver metastasis), prognosis, and anti-IGF2 therapy in CRC patients.

## 2. The IGF System in Carcinogenesis

IGF is a complex system, containing, apart from insulin, two IGFs (IGF1 and IGF2), their surface receptors, six insulin-like growth-factor-binding proteins (IGFBPs), IGFBP proteases, as well as insulin substrate proteins (IRS1-6). The functions of almost all of the elements of the IGF system were described in detail in numerous physiological processes (e.g., development, growth, aging, and neurological function) [95] as well as pathologies (e.g., growth deficiency, cancer, neurological, and cardiovascular diseases) [47,48,49,58,95,96,97]. The action of both IGF bases on a number of well-known signaling pathways common for many GFs (e.g., IGF1R and AKT1; mitogen-activated protein kinase (MAPK); the phosphatidylinositol 3-kinase (PI3K); and canonical and noncanonical β-catenin) [63,98,99,100] as well as those that were recently discovered (e.g., PLAGL2/IGF2/β-catenin) [101].

Production and secretion of different members of the IGF family in colon carcinogenesis are affected by both genetic and environmental factors [52,102,103]. Pathogenesis of CRC, also known as an obesity-associated cancer, was associated with overweight and obesity due to activation of the PI3K/AKT pathway [54,100]. Insulin and IGF signaling combined with chronic inflammation are also important factors in the CRC-promoting effects of obesity [104].

Numerous studies concerning the members of the IGF system in different tumor developments (including CRC) were mostly focused on IGF1, IGF1R, and IGFBP1-3 [47,74,97,105,106,107].

### IGF2 as an Enigmatic Component of IGF System—Role in Physiology

While IGF2, as foretold by René E. Humbel [108], belongs to the group of increasingly better characterised elements of multifactorial IGF signaling, it remains the relatively unknown element of the family. Particularly, the role of IGF2 requires more research focused on more specialized adult tissues as well as carcinogenesis (including colorectal cancer). These studies are complicated by the epigenetically controlled parental imprinting of H19/IGF2 genes, with disturbances resulting in genetic disorders including in vitro dysplasia and in vivo tumorigenicity [61,109]. When *IGF2* LOI occurs, the maternal allele may also be expressed, with some studies correlating *IGF2* LOI with increase in the expression of this gene [67].

IGF2, first isolated from human plasma fraction [110,111], together with IGF1 exhibited insulin-like activity as well as growth-promoting action on chick embryo fibroblasts (hence, the name insulin-like growth factor) [111]. Despite the facts that the primary structure of IGF2 was indicated [112], that its cDNA was isolated and described [113], and that the complexity of tissue-dependent *IGF2* transcription was discovered, biological functions of IGF2 have long been a matter of discussion. Data on this matter can be found in many excellent reviews [108,114,115].

Thanks to its most spectacular function, IGF2 is considered by many to be a mitogen [108], a major embryonic growth factor in mice and humans [114,116], a powerful driver of cell proliferation [63], and a major tumorigenic driver [91]. The *IGF2* gene was included in the group of pleiotropic genes, working as satellite regulators of the growth pathways [117]. In mice, IGF2 expression is halted in most of the tissues after birth but the gene is often reactivated during tumorigenesis [116]. Human *IGF2* gene expression and protein biosynthesis continues throughout life [118]. Together with IGF1 (under the control of GH), IGF2 is the dominant axis regulating postnatal somatic growth [64]. The discovery of *igf2* imprinting in mice, described by Chiara et al. in mice and later humans (reviewed in Reference [61]), began a new era in the way researchers look at IGF2 and its participation in carcinogenesis (including CRC). These aspects are the topic of further impressive reviews [58,61,74,103]. The most recent study of Barel and Rotwein adds new insights into *IGF2* gene structure and regulation, showing extensive conservation in coding regions of human/mice *IGF2/Igf2* exons and IGF2 proteins, with the presence of several moderately conserved 5′ untranslated (UTR) exons in *IGF2/Igf2* [118].

Mature human IGF2 protein is a small, acidic polypeptide consisting of 67 amino acids (aa) (7.5 kDa) and manifesting 65% homology with IGF1. It is expressed by parenchymal and non-parenchymal cells of liver and many other tissues [58,108]. Liver-synthesized IGF2 serves an endocrine function, while that produced by extra-hepatic tissues evokes autocrine/paracrine effects [108]. IGF2 action is tightly regulated and depends on a whole system of proteins, including those specific for IGFs—IGFBPs. IGF2, together with IGF1, IGFBP3, and acid-labile subunit (ALS), forms 150-kDa ternary complexes, present exclusively in circulation [119]. IGF2 is the predominant IGF in adult humans, with 3.5-fold higher serum levels than IGF1 [74,120]. Throughout life, it is constantly present in circulation at a substantially high level (400–700 ng/mL) [121].

Human *IGF2*, composed of 9 exons and located next to the insulin gene on the short arm of chromosome 11 (11p15.5), was first described in papers published in “Nature” [122,123]. Recent data extracted from public genomic and gene-expression databases indicates that *IGF2* of each primate, excluding those found in gibbons and marmosets, is composed of 10 exons and is regulated by five potential promoters, each with distinctive 5′-UTR exons [118,124]. *Igf2* in mice and *IGF2* in humans are tightly controlled by maternal imprinting, with only one allele expressed, depending on parental origin [64,65]. *IGF2* gene spans ~30 kb of chromosomal DNA [113,122].

In all fetal tissues, *IGF2* is transcribed from three ubiquitous promoters (P2–P4). In adult liver, *IGF2* gene transcription is initiated from a liver-specific promoter (P1) but the P2–P4 promoters remain active in adult peripheral tissues (reviewed in Reference [125]). It was proven that, while *IGF2* is imprinted in fetal liver, its biallelic expression occurs in the adult organ (*IGF2* transcripts from promoter P1 are always derived from both parental alleles) [65].

*IGF2* is translated as a 156 aa pro-ligand consisting of the N-terminal mature IGF2 sequence of 67 aa, together with a C-terminal 89 aa E-domain, which is glycosylated by up to four *O*-linked sugars [114,116,126]. IGF2 mRNA-binding proteins (IMPs/IGF2BPs) further regulate its translation [96]. Incomplete processing of pro-IGF2 results in various peptides (10–18 kDa), containing complete or partial E-domain and known collectively as “big” IGF2. The majority (71%) of total IGF2 in human plasma was the mature form, while “big” peptides and pro-IGF2 constituted to 16% and 13%, respectively, with more variation seen in the levels of mature IGF2 [120].

IGF2 signals mostly via three types of receptors: IGF1R, both insulin receptor’s isoforms (IR-A/B), and IGF2R/the cation-independent mannose-6-phosphate receptor (IGF2R/CI-M6PR); however, the pro-proliferative effects are exerted mainly through the IGF1R and IR-A [58,127,128]. IGF2 binds with high affinity (similar to that of insulin) to IR-A, amplifying mainly the mitogenic but not the metabolic effect [104,127] and may play a role in both fetal growth and cancer biology. Using IGF chimeras, it was proven that both flanks of the IGF2 C domain play important roles in the greater ability of the IGF2 to bind and activate IR receptors compared to IGF1 [129,130]. IGF2 also binds to the IGF-2R/CI-M6PR—an unrelated monomeric receptor that lacks tyrosine kinase activity and acts as a scavenger for circulating IGF2 [131,132,133]—and the IGF1R/IR-A hybrid receptor [58,128,134]. Molecular interactions between IGF2 and domain 11, one of 15 extracellular domains on IGF2R, were revealed [132]. Cell surface IGF2R is involved in clearing free IGF2 from the circulation by targeting it for lysosomal degradation [133]. It is worth adding that transcriptions of both *IGF2* and *IGF2R* are subject to genomic imprinting [135].

## 3. Known and Less Known Alterations of *IGF2* in Colorectal Carcinogenesis

### 3.1. Autocrine/Paracrine IGF2 Secretion

Autocrine/paracrine effects of IGF2 mainly concern the polypeptide produced in extra-hepatic tissues, including colorectal tumor cells and CRC tumor stroma (cancer-associated fibroblasts (CAFs)) [91,108,136]. Increases in tumor invasiveness, dissemination capacity, and local tumor regrowth were noted after inoculation of colon cancer cells with CAFs expressing endogenous IGF2 in mouse xenograft models. In addition, the expression of IGF2 correlates with elevated relapse and poor survival in CRC patients [136].

In vitro studies showed 80-fold increases in IGF2 production by tumorigenic clones of the SW613-S human colon carcinoma cell line compared to IGF1. Attention was also brought to the role of IGF2 as the main effector of the autocrine loop [137]. Additionally, functional studies confirmed that IGF2 overexpression plays a major role in the maintenance of tumorigenic phenotype of these cells [138]. Autocrine IGF2 constitutively activated IGF1R and AKT phosphorylation, which was inhibited by selective IGF1R/INSR inhibitor BI 885578 [139].

IGF2 overexpression is an accepted risk factor of CRC development [51,52,136]. In the autocrine loop functioning with increased IGF2 action, major roles are supposed to be played by alterations in the *IGF2* gene itself (e.g., LOI), binding to isoform A of the IR with stimulation of mitosis, and/or by impaired IR-A internalization and continued production of IGF2 [58,96].

### 3.2. Mechanisms of Increased IGF2 Gene/Protein Activities

There are several mechanisms responsible for the increase in *IGF2* gene/protein activity in human tumors. These are most often epigenetic changes, altered binding proteins, loss of transcriptional suppressor protein, activation of transcription factors, and/or altered *IGF2R* [51]. In the case of CRC, overactivation of IGF2 most commonly occurs through LOI epigenetic mechanisms as well as reactivation of the fetal *IGF2* promoters (especially P3) (reviewed in Reference [48]). LOI of *IGF2* is a frequent epigenetic change in CRC [28,67,68]: 63% CRC samples were found to have an LOI in tumors, while ~22% had an LOI in the normal control group [69]. Zhang et al. showed the LOI of *IGF2* in higher frequency than other authors, 87.5% in CRC and in ~70% of control tissue, but the study was based on a relatively small number of CRC (*n* = 34) and control tissues (*n* = 14) [86].

A pilot study indicated a significant relationship between the LOI of *IGF2* and the family as well as personal history of colorectal cancer [68]. In addition, LOI of *IGF2* is an early event in colorectal cancer development, activating IGF1R and AKT1 signaling pathways [63]. Vanaja et al. described the molecular basis of the *IGF2* LOI phenotype leading to increased cancer predisposition. Compared to wild-type cells, the same doses of IGF2 trigger a weaker activation of AKT1 and stronger activation of ERK1/2 signaling pathways in LOI cells, causing further rebalancing between pro- and antiapoptotic control mechanisms [28]. LOI of *IGF2* is also found in normal colonic mucosa (in the vicinity of tumors) of about 30% of CRC patients but only of 10% of healthy individuals [67]. However, LOI-independent mechanisms of *IGF2* overexpression were also described, particularly when transcription occurred at significantly high levels [140]. Zhong et al. proved that elevated *IGF2* expression in a subset of CRC cell lines is caused by an increase in DNA copy number and hypermethylation in the H19 promoter of the IGF2 gene [91]. An interesting study using tumor type-specific analyses uncovered that enhancer-hijacking is the main mechanism mediating the dysregulation of *IGF2* locus in CRC. A previously undescribed mechanism assumes tandem duplication-mediated de novo formation of a 3-D contact domain accompanying a super-enhancer normally inaccessible to *IGF2*, resulting in >250-fold gene upregulation [24].

There are also reports of changes in different receptors and signaling components expressed in cancer cells [52,58]. IGF2R, which sequesters free IGF2 ligands, can modulate IGF2 availability and its mitogenic activity [132,133]. While numerous genetic variants of *IGF2R* were described, the knowledge on their involvement in this receptor’s functioning, indirectly affecting IGF2 action, remains incomplete. Some of the data indicates that women homozygous for the *IGF2R* c.5002 G > A allele had significantly higher mean serum levels of mitogen IGF2 than noncarriers (828 ± 321 and 595 ± 217 ng/mL, respectively). Only homozygous individuals for the other *IGF2R* variant (c.901 C > G) trended towards a higher risk of CRC, with OR = 2.2 (95% CI (0.9–5.4)), although this association may not be via modulation of the circulating ligand [59].

In some primary and metastatic CRC patients, overexpression of IGF2 is accompanied by chromosomal gains at 11p15.5 [6,141]. A large-scale multi-institutional study proved that 7% of the patients’ primary tumors were characterized by a gain in the *IGF2* gene [6]. Another interesting matter is the presence of *IGF2* LOI in cancer stem cells isolated from all of the studied CRC cell lines (HT29, HRT18, and HCT116). This resulted in a higher rate of colony formation and greater resistance to chemotherapy and radiotherapy in vitro [142].

The precise mechanism(s) through which increased IGF2 induces colorectal carcinogenesis remain(s) largely elusive. However, there is some evidence that IGF2 may exert also its oncogenic function in CRC cells [91,96,143]. IGF2 is a direct, potent proliferation stimulator/co-stimulator for a variety of CRC cell lines, functioning as an autocrine growth factor [74,91,139,144,145]. Studies on the role of microRNAs (e.g., miR-483 and miR-486-5p) as tumor inducers in CRC confirm the involvement of the IGF2 pathway in CRC cell proliferation and invasion both in vitro and in vivo [101,143].

### 3.3. IGF2 Actions Relevant to CRC Development

A role of IGF2 in proliferation, cell cycle, apoptosis, cell adhesion, and stemness was noted in many functional systems [84,89,144,146,147]. Both IGFs are known as regulators of G1 to S phase transition [148]. In CRC, similarly to other cancers, IGF2 promotes cancer cell growth through several types of receptors, mainly IGF1R and AKT phosphorylation, as well as IR-A [58,139]. IGF2 similarly to IGF1 co-activates proliferative and apoptotic pathways in LIM 1215 colon cancer cells, which may contribute to increased cell turnover [146]. In vitro studies also showed that IGF2 can play an important role in cell proliferation as well as differentiation of “normal” colonic cells [149] and CaCo-2 colon carcinoma cell line [89].

IGF2 action results in induction of activated and repressed genes (e.g., *Wnt5a*, *CEACAM6, IGFBP3*, *KPN2A*, *BRCA2*, and *CDK1*) [63]. Rogers et al. showed that a knockdown of *IGF2* decreases proliferation and IGF2 mRNA production in two CRC cell lines (invasive and nonmetastatic SW480 and metastatic LS174T cell lines). A deregulation of 58 genes was indicated by the microarray analysis of *IGF2* knockdown SW480 cells, with several of them associated with proliferation and cell–cell/cell–ECM contact. G1 phase blockage of the cell cycle was associated with *IGF2* knockdown. Hence, the concept of direct autocrine/paracrine tumor cell activation through IGF2 is supported by this study [147].

In vivo studies also note that IGF2 tissue expression in CRC correlates with expression of other proliferation markers, e.g., proliferating cell nuclear antigen (PCNA), which confirms the role of this growth factor in tumor cell proliferation through a paracrine mechanism [84,92]. Recent studies based on TCGA analysis revealed that, in human colon cancer specimens, a stronger activity of β-catenin/TCF responsive promoter was associated with a higher transcription of *IGF2* and *IGF1R*, which can be important in design of therapies employing IGF1R/IR inhibitors [150].

### 3.4. Epidemiological Evidence of Circulating IGF2 Association with CRC Risk

Studies on the association of circulating IGF2 with the risk of CRC development deliver mixed results. Older literature positions note that the circulating IGF2 levels are unrelated to the risk of CRC [107,151]. Other studies indicate that individuals with IGF1 and IGF2 values in the upper 2 terciles of the respective distributions had a significantly elevated CRC odds ratio (OR = 5.2, 95%; CI 1.0–26.8) compared with those in the lower tercile of both distributions [152]. Newer literature contains studies describing higher serum IGF2 concentrations in CRC compared to control [77], male-specific higher concentrations in CRC [78], as well as CRC concentrations comparable to control [80]. In a study focused on oriental patients, a positive correlation between the concentrations of IGF2, IGFBP3, and CRC risk was noted when cases were confined to those diagnosed within a relatively short time period after enrollment (within 8 years). According to the authors, both of these components of the IGF system (including IGF2) can serve as early indicators of impending colorectal cancer [153]. A positive correlation is noted between IGF2 concentration and more advanced CRC [79,80] as well as a tendency for such correlation with disease stage increase and metastases to regional lymph nodes [78]. In contrast, Liou et al. showed that higher IGF2 plasma levels were associated with reduced risk of mortality [80]. Higher mean IGF2 SD scores compared with controls were observed in Dukes A and Dukes B patients but not in those in advanced stages of the disease [154]. Elevated serum IGF2 concentrations were more often described in patients with *IGF2* LOI, always being an indicator of bad prognosis [80].

The only available meta-analysis of 19 epidemiological studies (5155 cases and 9420 controls) showed that high IGF1 and IGF2 levels significantly increased CRC risk (OR = 1.52, 95% CI: 1.16–2.01 for IGF2) [81].

Summarizing, the epidemiologic data on IGF2 serum levels and CRC mostly showed an association between elevated serum IGF2 levels and increased CRC risk. However, the role of serum IGF2 as a predictor of CRC progression should be better characterized. The available data also suggests that evaluation of circulating IGF2, together with the epigenetic alterations (e.g., *IGF2* LOI), might be useful to assess the severity of the CRC and could be a target for a novel form of therapeutic strategy in CRC.

### 3.5. Tissue Expression of IGF2 in Primary and Metastatic CRC

Subcellular IGF2 localization (mRNA and protein) in CRC patients mainly concerns the cytoplasm of neoplastic cells of both primary and metastatic tumors. IGF2 immunostaining was also observed in normal liver adjacent to tumor masses [60,84,93]. The mainly cytoplasmic IGF2 localization in CRC was confirmed, with the use of immunohistochemistry, in our own studies (unpublished data) (Figure 1).

Quantitative studies show an increased tissue IGF2 expression (mRNA and protein) in a number of colonic cancer cell lines derived from adult tumors [84,88,91,149] as well as in human primary and metastatic colonic carcinoma [82,83,84,85,87,93,94,140]. There are some studies that did not show any quantitative differences in IGF2 mRNA expression between cancer tissues and adjacent normal mucosa [155,156]. Comparison of expression studies of mRNA levels of all the IGF-system components (including IGF2) in normal colorectal tissues showed significantly higher levels in the rectum compared to the ascending colon [157]. Interestingly, higher IGF2 expression—above 10 to 50-fold [82,83,85,87]—or a significant increase in IGF2 mRNA production (200–800-fold) [83,87] concerns only a part of the studied CRC patients, e.g., ~33% [83], ~38% [60], and ~40% [82]. According to some of the researchers, IGF2 is the most differentially expressed gene between carcinoma and adenoma lesions [60]. The definition of IGF2 overexpression described by Sanderson et al. (2017) is based on a comparison of differential expression between cancerous and normal tissue samples. Based on two separate datasets, commoness of IGF2 overexpression was noted in 13–22% of these tumors [139]. A generally higher expression of IGF2 mRNA (40-fold) was observed in comparison with the protein itself (2-fold) [85]. There is a discussion on the prognostic effect of IGF1 and IGF2 expression in CRC, which seems to be of limited value if a multivariate (but not in univariate) survival analysis is performed [94]. However, a general detection of IGF2 expression/overexpression in primary CRC and liver metastases is most commonly linked to disease progression, grade or stage increase, or worse survival prognosis [84,92].

The role of IGF2 in the CRC metastasis process remains an open matter [5,10,25,101]. Studies conducted on highly metastatic CRC cell lines point towards IGF1 and IGFBP1 participation rather than IGF2 as potential genes involved in CRC metastasis [158]. There are also some results considering IGF2 as an important tissue marker in tumor progression in CRC with liver metastases [92,93]. Additionally, expression of only three proteins, namely TGF-α, IGF2, and MMP-2, was significantly elevated in the metastatic CRC group, independently of the other variables (e.g., tumor classification, histological grade, and patient age) [93]. Singular studies indicate lower expression of IGF2 in metastases compared to CRC and/or control [87]. A list of genes and proteins related to liver metastases formation was also compiled, with no *IGF2* present in that group yet [4,5,20,158,159].

Other current studies apply *IGF2* in a panel with other CRC-specific markers (e.g., *CACNA1G*, *CDKN2A*, *CRABP1*, *IGF2, MLH1*, *NEUROG1*, *RUNX3*, and *SOCS1*) to study the CpG island methylator phenotype (CIMP) phenotype in primary and metastatic CRC [25,70,160]. Recent study showed higher CIMP(+) frequency in patients diagnosed with multiple CRC than in patients with “unique” CRC [161]. It was proven that CIMP-high cancers had poor survival outcomes in CRC patients [160]. In turn, when it comes to CRC metastasis, CIMP status is generally concordant between primary CRCs and corresponding metastases [25]. Using paraffin-embedded CRC tissue samples, it was proven that hypomethylation of the *IGF2* DMR0 was significantly associated with higher overall mortality [162]. During interpretation of the results of studies on the involvement of various genomic aberrations, allelic imbalance at cardinal loci, and CIMP in CRC, several groups of genomic aberrations are pointed out (mainly “insular” genomic aberrations), of known or still unclear role in CRC. Some of these changes might hold importance in clinically metastasizing disease [10].

### 3.6. Non-coding RNAs Regulated by IGF2 in CRC

Recent research indicates dysregulation of different mRNAs, micro- and long non-coding RNAs (miRNAs and lncRNAs, respectively), as critical modulators of the CRC progression [15,35,39,163,164,165,166,167]. The lncRNAs are a major type of non-coding RNAs (ncRNAs), playing a role in many biological processes in CRC (reviewed in Reference [30]), whereas the miRNAs are short, single-stranded ncRNA sequences of ~21–23 nucleotides, with the expression unique for different tissues (including tumor tissue). These molecules participate in maintaining differentiated cellular states, with disorders of miRNA expression able to shift cells into the undifferentiated proliferative phenotype [167].

Due to the large abundance of data, this section bases only on the studies concerning the *IGF2* gene and/or pathway associated with different lncRNAs and miRNAs as a diagnostic-prognostic biomarker or therapeutic target of CRC.

The first report of a lncRNA regulated by insulin/IGFs was published by Ellis et al. [37]. The authors’ findings indicate a role of *CRNDE* (colorectal neoplasia differentially expressed) nuclear transcripts (lncRNAs) in regulating cellular metabolism, which may correlate with their upregulation in CRC. Current meta-analysis of cancers of the digestive system (including CRC) indicates that lncRNA CRNDE could predict worse prognosis in CRC [166].

One of the fascinating lncRNAs linked with *IGF2* is the H19 lncRNA. Human *H19* gene was cloned and sequenced by Brannan et al. in the “pre-genomic” era [168]. *H19* and *IGF2* are two oppositely expressed genes, located on the same chromosome (11p15.5), sharing the same transcriptional regulatory epigenetic mechanisms [169]. *H19* was the first imprinted non-coding RNA transcript identified, with the *H19/IGF2* locus (known CCCTC-Binding Factor (CTCF)-binding element) serving as a paradigm for the study of genomic imprinting since its discovery [35,170]. The H19 lncRNA seems to have an important role in colonic carcinogenesis, proven by increasingly numerous studies [40,46,163,165]. Hu et al. noted that the lncH19 RNA was significantly increased in immunodeficient mice induced with human colon cancer cells compared to controls [163]. In turn, Yang et al. proved that the H19 lncRNA was overexpressed in colon cancer tissues and cell lines, while the short hairpin RNA (shRNA) interference of H19 effectively decreased the migration and invasion of colon cancer cells (HT-29 and RKO). The authors implicated the H19-miR-138-HMGA1 pathway in regulating the migration and invasion of colon cancer [165].

There is a discussion on the correlations between the degree of DNA methylation (hypo- and hypermethylation) of imprinting-associated *IGF2/H19* domain DMR and *IGF2* LOI in colorectal cancer and normal colon mucosa [69,140,171,172,173,174]. Nakagawa et al. reported that *IGF2* LOI correlated strongly with biallelic hypermethylation of the core five CpG sites in the insulator region of *IGF2/H19* in both tumors and normal colonic mucosa [171]. Another study indicates a relation between *IGF2* LOI and *IGF2*-DMR0 hypomethylation but not *H19*-DMR hypermethylation in CRC. These authors indicated that this epigenetic abnormality was not limited to cancer tissues, with the same hypomethylation pattern in normal colonic mucosa and tumors of each patient [172]. Cheng et al. have shown a significant correlation between *IGF2* LOI and DMR hypomethylation of *IGF2* and *H19* but that there was no correlation between marked IGF2 elevation and *IGF2* or *H19* hypomethylation [140], whereas Tian et al. showed that, in primary CRC, *IGF2* LOI is associated with hypomethylation of the six CTCF-binding sites in the *IGF2/H19* DMR. No correlations with clinical variables were found for *IGF2* LOI, suggesting that the incidence of *IGF2* LOI is an early event in cancer progression [69]. Recent studies by Hidaka et al. showed that, among four imprinting-associated DMRs, the *IGF2*-DMR0 hypomethylation was not associated with *IGF2* LOI but occurred most frequently within the *IGF2/H19* domain [174].

Research points to other possible causes of *IGF2* LOI than DNA methylation within the *IGF2/H19* domain, e.g., sequence alteration of *H19*-DMR, dysfunction of the intrachromosomal loop, or expression of the CTCF or PRC2 complexes [173]. Hence, it seems that the mechanism for the role of aberrant methylation of imprinting-associated DMRs in mediating outcome of CRC is still unknown [174].

Han et al. implicated H19 lncRNA upregulation in CRC tissues in high Tumor-Node-Metastasis (TNM) stage and poor differentiation [175]. Similarly, elevated expression of *H19* lncRNA in cells isolated from metastases due to promoter demethylation was associated with poor survival of colon cancer patients [38]. It was also proven that *H19* and *miR-675* (processed form of the former) were found to be upregulated in human colon cancer, both in cell lines and primary CRC biopsies, compared to adjacent noncancerous tissues [176].

In other functional analysis, the key lncRNAs in the competing endogenous RNA (ceRNA) network were noted. The group of three lncRNAs for which a link with certain clinical features (i.e., distant metastases) was found included the *IGF2* antisense (*IGF2*-AS), long intergenic non-protein coding RNA 00355 (*LINC00355*), and hepatocellular carcinoma up-regulated lncRNA (*HULC*). This group of researchers also showed that specific miRNAs may target lncRNA IGF2-AS (e.g., has-mir-98, -152, -182, and -424) [15]. Liang et al., using the Kaplan–Meier curves, showed that the *IGF2*-AS lncRNA and four other DEGs (POU6F2-AS2, hsa-miR-32, hsa-miR-141, and SERPINE1) were negatively correlated with OS in CRC patients [34].

Currently (2019), in early- and late-stage CRC, almost 550 common differentially expressed mRNAs and 30 lncRNAs were identified after comparison with normal colon samples. Among them, the authors found that the lncRNA H19 was significantly upregulated in CRC tissues compared with adjacent normal control. The authors pointed out that upregulation of *H19* might regulate the expression of FSCN1—an actin-binding protein—by competitively sponging miR-29b-3p [40]. Dai et al. identified 4307 lncRNAs and 5798 mRNAs deregulated during CRC initiation. Among them, lncRNA H19 was upregulated in colon tumors and correlated with poor patient prognosis [46].

There are some interesting research results on the role and mechanisms of action of microRNA (miRNA and miR) via IGF2 in CRC [101,109,143,177,178]. The model of colon organoid culture functionally validated the miR-483 as a dominant driver oncogene at the *IGF2* 11p15.5 CRC amplicon, inducing neoplastic colon transformation [109]. Cui et al., in a study on Hsa-miR-483 (located within intron 7 of the *IGF2* locus), described simultaneous increases in the levels of IGF2, miR-483-3p, and miR-483-5p in CRC tissues [143]. There are also some interesting results of a study concerning regulatory mechanisms of certain miRNAs in CRC. It was showed that a decrease in miR-486-5p levels was associated with larger tumor size, advanced TNM stage, poor CRC prognosis, and lymphatic metastasis. In turn, miR-486-5p promotion of CRC proliferation and migration, mediated by the activation of PLAGL2/IGF2/β-catenin, was indicated by bioinformatical analysis. Significantly upregulated plasma miR-486-5p expression was also noted in CRC patients [101]. Furthermore, the most recent research by Lu et al. point towards IGF2 as a direct “target” of another miRNA, namely miR-491-5p, in CRC cells. Overexpression of IGF2 rescued the miR-491-5p-induced suppression of CRC cell proliferation. Additionally, these authors noted a decreased plasma miR-491-5p expression in CRC compared to healthy controls [177]. The authors imply miR-491-5p as a CRC suppressor via IGF2 targeting. They suggest that this microRNA can be used as a diagnostic-prognostic biomarker in this cancer type.

Recent data on three other lncRNAs (miR-181a/135a/302c) showed that all of them function as tumor suppressors, repressing PLAG1/IGF2 signaling, and exert their effect on 5-fluorouracil (5-FU) chemoresistance through attenuation of PLAG1 expression [178].

The main genetic and epigenetic changes of *IGF2* and the potential clinical significance of IGF2-associated ncRNAs (lncRNAs and miRNAs) in CRC are summarized in Table 1.

In Figure 2, we summarized the most important genetic and epigenetic alterations of the IGF2 gene, IGF2-associated non-coding RNAs, as well as gene-regulated mechanisms and signaling pathways in colorectal carcinogenesis.

## 4. New Targets in Anti-IGF2 Colorectal Cancer Therapy

Therapeutic targeting of IGF signaling has been investigated in many types of human cancer (including colorectal cancer) [52,179]. A publication by Simpson et al. [179], as well as many other excellent reviews, states that the insulin/IGF system is implicated in the development of resistance to both chemotherapeutic drugs and EGFR-targeted agents. The IGF2 gene itself is associated with aggressive and chemotherapy resistant colorectal cancer [103]. Increased expression of IGF2 is linked to early relapse and decreased progression free survival in CRC cases. Also, its increased expression was found in the tumor samples from patients who have been pretreated with radiotherapy [136,180]. Hence, complete understanding of the biological IGF2 functions in the context of the whole IGF system is the only condition that needs to be met before this protein can be used as a potential “target” in CRC therapy. While there are some reports on the use and effects of insulin/IGF-targeting trials in colorectal cancer (reviewed in References [104,179]), there is not much published data regarding the effect of the CRC treatment using precise anti-IGF2 approaches. Some of the results of cell line or xenograft studies are presented below.

### 4.1. Cell Line Research

Recent data notes that autocrine IGF2 overexpression intensifies activation of IGF1R phosphorylation followed by AKT signaling, which can be inhibited by selective IGF1R/INSR TKI (tyrosine kinase inhibitor) BI 885578. It was shown that IGF2 expression was the feature most significantly associated with BI 885578 sensitivity. In a large panel of CRC cell line constitutive basal IGF1R phosphorylation studies, combined with an analysis BI 885578 sensitivity, hyperactivation of the IGF2-driven pathway was only observed in cell lines characterized by IGF2 expression larger than 250 TPM (transcripts per million). Abundant basal phospo-IGF1R signals were decreased by over 90% in BI 885578 treated cells. Additionally, several non-CRC cell lines sensitive to BI 885578 were characterized by low IGF2 expression and basal phospho-IGF1R levels. The authors pointed out that the knowledge about identification of common sensitivity determinants in different cell lines could be useful for studies on the utilization of IGF pathway inhibitors [139].

CRC gene therapy is yet another novel idea, with the recombinant oncolytic virus targeting the *IGF2* imprinting system indicated as a potential new anticancer agent [181,182]. The study of Nie et al., newly employing the Ad315-E1A oncolytic adenovirus and the Ad315-EGFP replication-deficient recombinant adenovirus, driven by the *H19* promoter inserting *IGF2* imprinting system, discovered that Ad315-E1A significantly impaired viability of the cells and only induced apoptosis in in vitro cultured *IGF2* LOI cell lines [181]. Another study of the same group showed that adenovirus-mediated siRNA targeting CD147, rAd-H19-CD147mirsh, and driven by the same *IGF2* imprinting system used in the *IGF2* LOI cell lines also decreased cell viability and invasiveness as well as increased sensitivity to chemotherapeutic drugs in these cell lines [182]. Previous data from the same authors showed that infection with Ad-DT-A resulted in growth inhibition (75.4 ± 6.4%) and increased the percentage of apoptosis (20.8 ± 5.9%) in human CRC HCT-8 cell line (*IGF2* LOI) compared with control group [183]. Others proved that infection of *IGF2* LOI cell lines with Ad312-E1A reduced cell viability and induced their apoptosis [184].

### 4.2. CRC Xenograft Model Research

Sanderson et al. examined the in vivo efficacy of BI 885578 in IGF2-high and IGF2-low CRC cell lines xenografted to mice. During BI 885578 oral therapy, they observed significant inhibition of tumor growth in the IGF2-high models, with tumor growth inhibition values between 54–92%. Inversely, in IGF2-low model treatment, BI 885578 did not significantly inhibit tumor growth. Also, the combination of BI 885578 with a Vascular endothelial growth factor A (VEGF-A) antibody increased treatment efficacy and induced tumor regression [139]. Other authors decided to regulate the bioavailability of IGF2 at the protein level using the MEDI-573 IGF1/2 neutralizing antibody. They treated CRC mouse models with high expression of IGF2 using MEDI-573. This therapy resulted in significant tumor growth inhibition. It was also noted that IGF2 inhibition by MEDI-573 modulated other signaling pathways. Hence, combination therapy employing such inhibitors as trastuzumab (HER2 monoclonal antibody), AZD2014 (dual mTORC1/2i), AZD5363 (AKTi), and selumetinib (AZD6244/ARRY-142886, MEK1/2i) or cetuximab (anti-EGFR agent) significantly enhanced in vivo efficacy of the drug [91]. However, it should be emphasized that the antibodies which usually bind to IGF2 to form complexes are absorbed and degraded, showing single-use efficacy. This indicates that singular administration of such antibodies may not be effective enough, requiring further studies in human therapies.

Zanell et al., in an interesting study of CRC samples from “xeno-patients” responsive to cetuximab, showed that a subset of cases (37.5%) in which enhanced EGFR inhibition was unproductive exhibited marked overexpression of IGF2. In addition, in functional studies, IGF2 overproduction attenuated the efficacy of cetuximab [185]. The authors suggest the use of combined CRC therapies and provide evidence for the role of IGF2 as a biomarker of reduced tumor sensitivity to anti-EGFR therapy as well as a determinant of response to combined IGF2/EGFR targeting. Other research, analyzing patient-derived tumor xenografts and response to cetuximab, noted frequent IGF2 upregulation (16%) that was mutually exclusive with IRS2, PIK3CA, PTEN, and INPP4B level alterations, supporting IGF2 as a potential drug target [186].

Another therapeutic approach is the use of a transgene expressing a soluble form of IGF2R/CI-M6PR (sIGF2R) in the intestine. This specific inhibitor of IGF2 bioavailability, when administered to an IGF2 overexpressing colon cancer mouse model, rescued Igf2-dependent intestinal and adenoma phenotype [187].

Gene therapy was also examined in mice bearing xenografted tumors. Mice implanted with HCT-8 tumors and treated by intratumoral administration of the Ad315-E1A oncolytic adenovirus showed significantly reduced tumor growth and enhanced survival [181]. After intratumoral administration of the rAd-H19-CD147mirsh, significant reduction of the tumor growth and enhanced mice survival were observed [182]. The other adenovirus vector, Ad312-E1A, suppressed tumor growth in nude mice xenografts [184]. Infection of HCT-8 tumor-bearing nude mice with recombinant adenoviruses (Ad-DT-A) resulted in inhibition of the tumor growth with the rate of 36.4% [183].

Until now, there is no published data on the use of the therapeutic approaches described above in humans.

## 5. Concluding Remarks and Future Directions

The *IGF2* gene belongs to the most complexly regulated growth factors known. It is also the first imprinted gene displaying loss of imprinting (LOI) or aberrant imprinting in human cancers. In physiological environment, IGF2 usually supports normal cellular growth. Discovery of a connection between *IGF2* LOI with colorectal cancer was a major success, with effects of this defect including *IGF2* overexpression and, moreover, global chromatin instability, increased cell proliferation, and tumor development. This epigenetic alteration was even described as a “manifestation or biomarker of colorectal cancer”, which means that detection of *IGF2* loss of imprinting could serve as a valuable diagnostic tool. However, it needs to be noted that, in CRC, LOI of *IGF2* occurs in ~30–88% of tumors and ~20–70% of noncancerous tissues [67,140].

Genetic and epigenetic changes of *IGF2* and related IGF system genes during CRC progression, leading to altered gene regulation, seem to be determinant in the development of this malignancy. These changes mostly include DNA hypomethylation and cases of both aberrantly increased and decreased methylation of the IGF2 gene and associated genes (*H19*). The lowest degree of methylation was found to be present in cells isolated from metastases [38,172]. A mechanism typical for LOI, with hypermethylation of a DMR upstream of the H19 gene allowing for activation of the normally silent maternal *IGF2* allele, does not apply to CRC. It has been shown that *IGF2* and *H19* DMRs in CRC do not follow the reciprocal methylation pattern, suggesting association of *IGF2* LOI in CRC with *H19* and *IGF2* DMR hypomethylation [172].

The next major challenge is the discovery of molecular mechanisms of high-level IGF2 gene activation, since the abberrant *IGF2* expression in CRC consists of at least LOI, most probably combined with other mechanisms resulting in elevated *IGF2* expression [140].

Currently, numerous studies employing bioinformatical analyses concern the discovery of molecular biomarkers of different stages of colon carcinogenesis as well as potential “targets” of directed therapies [15,34,175,177,178,188]. Such studies indicate dysregulation of different long non-coding and microRNAs (lncRNAs and miRNAs, respectively) in CRC carcinogenesis, with some of them indicated as potential diagnostic, prognostic, and therapeutic CRC biomarkers (e.g., lncH19 RNA, H19-derived miR-675, H19-miR-138-HMGA1) [15,165,175,176]. There are reports of miRNAs that may target particular lncRNAs of the IGF2 system (e.g., *IGF2*-AS) as well as specific miRNA interactions with mRNAs (e.g., IGF2BP1) [15]. The potential and commercially available DNA methylation CRC biomarkers (2013) are summarized in some excellent reviews [188]. From the IGF family, only *IGFBP7* is included in this list as a potential biomarker in early CRC detection (stage I) and prognosis (stage III) [188]. Similarily to *IGF2,* a potential DNA methylation biomarker, *IGFBP7* methylation, has also been linked to CIMP-high CRC. A recent study also showed that multiple CIMP marker genes (including *IGF2*) showed significantly increased methylation in DNA mismatch repair proficient (MMR-P) adenomas of Lynch syndrome patients [189].

In general, the molecular mechanisms of epigenetic alterations of *IGF2* and associated genes are still unclear. On one hand, *IGF2* DMR hypomethylation may activate alternative transcription of suppressed genes, while on the other hand, gene hypermethylation may promote carcinogenesis through oncogene activation. When affecting promoters, CpG island hypermethylation leads to gene silencing and promotes tumorigenesis through well-known functional pathways.

Summarizing, studies of the IGF2 system in CRC over the last couple decades, including the evaluation of genetic/epigenetic *IGF2* changes, allowed for the following:

(1) More precise definition of the role of IGF2 (overproduction, activation of various signaling pathways, and change of gene expression) involvement in CRC pathogenesis;

(2) Determination of CRIS-D subtype of CRC (with *IGF2* overexpression and amplification) for a superior prognostic assessment of CRC patients and the role of IGF2 in desensitization to EGFR blockade in patients with *KRAS* wt tumors;

(3) Recognition of IGF2 as an important and selective marker of CRC progression and staging;

(4) Indication of *IGF2* as one of the panels of several CIMP-specific markers (together with *CACNA1G*, *NEUROG1*, *RUNX3*, *SOCS1*, *CDKN2A*, *CRABP1*, and *MLH1*) in diagnostics of chromosomal instability (CIN) or CpG island methylator phenotype (CIMP) in colorectal tumors;

(5) Inclusion of IGF2 as a candidate target for alternative treatment protocols or combinatorial anticancer therapy;

(6) Selection of IGF2 and its related molecules for use in CRC screening, diagnosis, prognosis, and therapeutic outcomes;

(7) The complexity and intertumoral variability of CRC probably hinders the progress towards developing IGF2-targeted therapeutics and will require a more accurate determination of the molecular profile of IGF2-sensitive tumors.

In the future, it is important to more accurately identify the role of the IGF2 in the mechanisms of metastasis and resistance to therapies that target the components of the IGF2 system in CRC. Additionally, the advisability of the use of anti-IGF2 in colorectal cancer in vivo needs to be determined. Further studies are required to determine which CRC patients are most likely to benefit from anti-IGF system therapy.

## Figures and Tables

**Figure 1 ijms-20-04915-f001:**
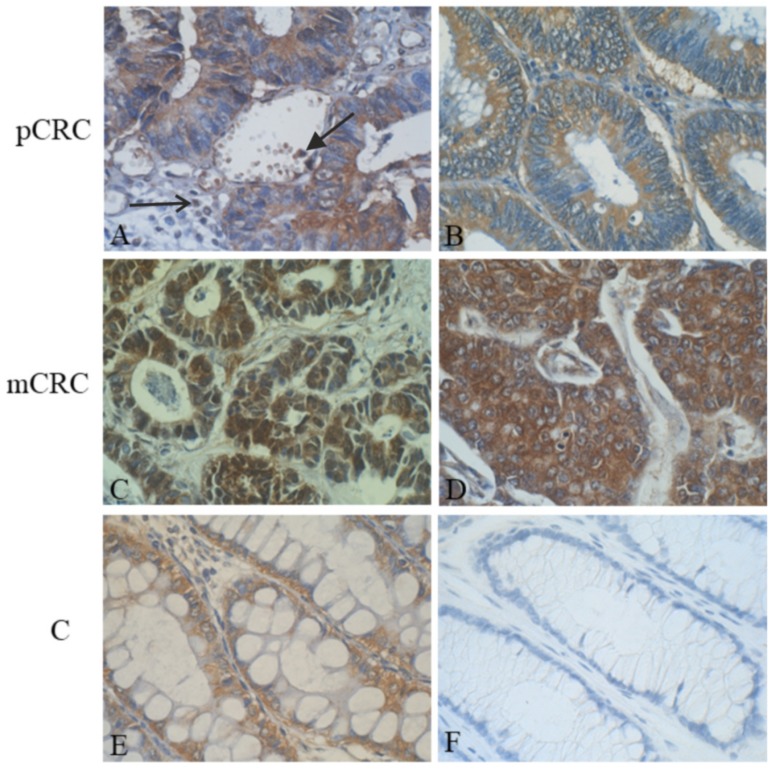
Immunohistochemical (IHC) localization of insulin-like growth factor (IGF) 2 in primary colorectal cancer (pCRC), metastatic colorectal cancer (mCRC), and control colon (C) samples: (**A**) A representative IHC expression of IGF2 mainly in cytoplasm of surface epithelium of tumor-changed colon crypts, in few white blood cells (arrowhead), and in individual cells of the tumor stroma (arrow); (**B**) homogenous cytoplasmic IHC reaction in neoplastic cells lining the glandular structures of colorectal cancer (CRC); (**C**) an intense IHC reaction of IGF2 in the cytoplasm and nuclei of numerous cancer cells present in lymph node metastatic carcinoma; (**D**) very strong cytoplasmic IHC reaction of IGF2 in cancer cells of lymph node metastatic CRC of other patient; (**E**) cytoplasmic expression of IGF2 in majority of goblet cells in normal colon epithelium; and (**F**) negative IgG control. New polymer-based immunohistochemistry with 3,3′-Diaminobenzidine (DAB) was the chromogen. Hematoxylin counterstained. Objective ×40 (Figure 1**A**–**F**) (our unpublished data).

**Figure 2 ijms-20-04915-f002:**
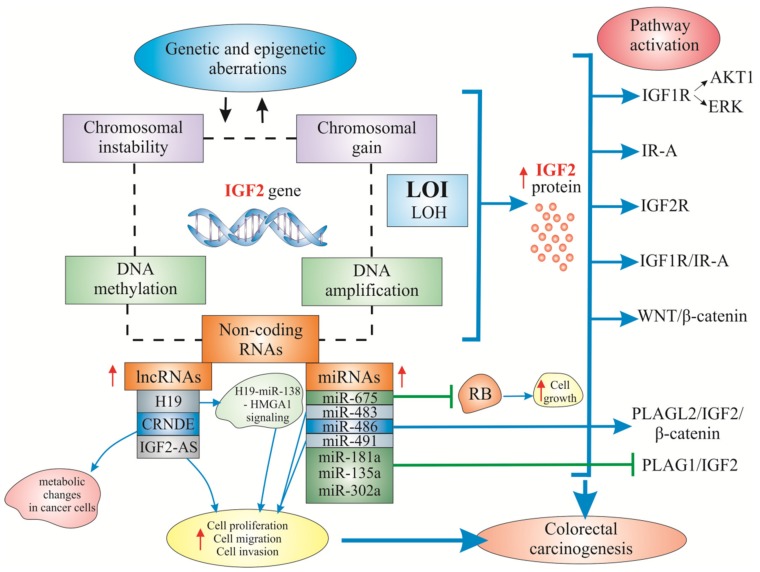
IGF2 gene and protein-associated non-coding RNA-regulatory mechanisms and the best-known IGF2-associated signaling pathways in colorectal carcinogenesis. Legend: ⇓ regulation; ↑/↓ increase/decrease; ⊥: inhibition.

**Table 1 ijms-20-04915-t001:** Summary of the main genetic and epigenetic changes of *IGF2* and IGF2-associated biomarkers in CRC.

Biomarker	Type of Change	Summary of the Findings	Potential Clinical Significance
*IGF2* gene	**LOI**	The adjusted OR for *IGF2* LOI in lymphocytes was 5.15 for patients with a positive family history, 3.46 for patients with adenomas, and 21.7 for patients with CRC [67].	Diagnosis
		*IGF2* LOI occurs not only in colon cancer tissues but also in matched normal tissues and peripheral blood cells [68].	Screening and diagnosis
		*IGF2* LOI were associated with increased risk of mortality in patients with stage IV disease; higher plasma IGF2 levels were associated with reduced risk of mortality [80].	Diagnosis and prognosis
		*IGF2* LOI was in 63% of CRC and in 21.7% of the normal control group (*p* < 0.01) [69].	Screening and diagnosis
		*IGF2* LOI with IGF2 overexpression [63]	Screening and diagnosis
		*IGF2* LOI was a common feature in CSCs; increased IGF2 was found in CSCs isolated from CRC cell lines (HT29, HRT18, and HCT116); higher rate of colony formation and greater resistance to chemotherapy and radiotherapy were found in vitro [142].	Diagnosis, mechanisms, and therapeutics
		*IGF2* LOI leads to rebalancing of activities of canonical AKT and ERK pathways; altered signaling balance leads to rebalancing of pro- and antiapoptotic control [28].	Diagnosis, mechanisms, and therapeutics
	**Amplification**	Colorectal intrinsic D (CRIS-D) subtype was specifically enriched for amplification of chromosome 11p15.5 and *IGF2* high-level overexpression in CRC [17].	Diagnosis and mechanisms
	**DNA methylation**	Biallelic hypermethylation of a core of five CpG sites in the insulator region of *IGF2/H19* correlated strongly with *IGF2* LOI [171].	Diagnosis and mechanisms
		Hypomethylation of the *H19* differentially methylated region (DMR) and DMR upstream of exon 3 of *IGF2* was found in CRC and normal colon mucosa; normal imprinting in the colon and LOI in CRC is specifically linked to the methylation status of a DMR within *IGF2* and not *H19* [172].	Diagnosis and mechanisms
		DMR *IGF2/H19* hypomethylation correlated with *IGF2* LOI; IGF2 overexpression did not correlate with *IGF2/H19* hypomethylation but negatively correlated with MSI [140].	Diagnosis and mechanisms
		Lower levels of *IGF2* DMR0 methylation were found in CRC than in control mucosa; *IGF2* DMR0 hypomethylation associated with male sex, low tumor grade, microsatellite instability (MSI)-low/microsatellite stable (MSS), CpG island methylator phenotype (CIMP)-low/0, wild-type proto-oncogene B-Raf (wt BRAF), and proto-oncogene K-ras (KRAS) mutation and higher overall mortality [162].	Diagnosis and prognosis
		Hypomethylation of the six CTCF-binding sites in *IGF2/H19* DMR were linked to *IGF2* LOI (two forms of aberrant IGF2 expression) and promotes MSI and oncogenesis. [69]	Diagnosis and mechanisms
		*IGF2* was between the five genes with the highest average hypermethylated percentages (50.4%) of CRC patients [56].	Diagnosis
		Lower levels of *IGF2* DMR0 methylation and ≥15 g/day alcohol consumption were associated with elevated risk of CRC [71].	Diagnosis and mechanisms
		*IGF2* DMR0 hypomethylation and aberrant methylation of other iDMRs within the IGF2/H19 domain with no association with *IGF2* LOI [174]	Diagnosis and mechanisms
		The “insular” genomic aberrations in *IGF2*; mosaic distribution of methylation in 1 region of the primaries or the metastases [10]	Diagnosis and prognosis
	**Chromosomal gains**	IGF2 overexpression in primary CRC and liver metastases was accompanied by chromosomal gains at 11p15.5 in a subset of CRC patients [6,141].	Diagnosis and prognosis
	**LOH**	LOH of *IGF2* was present in 13% of CRC patients with 33% with LOI of the IGF2 gene [70].	Diagnosis
	**Others**	A tumor type-specific analysis uncovered that enhancer hijacking mediates gene dysregulation at the *IGF2* locus in CRC [24].	Mechanisms
*IGF2R* gene	**Genetic variants**	Women homozygous for the *IGF2R* c.5002 G > A allele had higher mean levels of sIGF2; Whites homozygous for *IGF2R* c.901 C > G variant had a higher risk of CRC [59].	Diagnosis and prognosis
lncRNAs	***IGF2*-AS**	This key type of lncRNAs has correlation with certain clinical features (e.g., negative correlation between upregulation of IGF2-AS and distant metastasis) [15].	Diagnosis and prognosis
		This type of differentially expressed genes (DEG) was negatively correlated with overall survival (OS) [34].	Prognosis
	***H19***	This type of lncRNA was increased significantly in immunodeficient mice induced with human colon cancer cells when compared with controls [163].	Prognosis and therapeutics
		H19 upregulated a series of cell-cycle genes; eIF4A3 binds to H19. Higher expression of H19 was correlated with tumor differentiation and advanced Tumor-Node-Metastasis (TNM) stage [175].	Prognosis, mechanisms, and therapeutics
		H19 overexpressed in CRC tissues and cell lines; the interference of H19 by shRNA effectively decreased the migration and invasion of CRC cells. H19 shRNA strongly reduced the tumor growth and tumor volume in vivo [165].	Mechanisms and therapeutics
		Elevated expression of *H19* lncRNA due to promoter demethylation was observed in cells isolated from metastases and was associated with poor survival of CRC patients [15].	Diagnosis and prognosis
		H19 was significantly upregulated in CRC tissues compared with normal control, which might regulate FSCN1 expression by competitively sponging miR-29b-3p [40].	Diagnosis and mechanisms
		H19 was upregulated in colon tumors and correlated with poor prognosis [46].	Diagnosis, mechanisms, and prognosis
	***CRNDE***	Regulates cellular metabolism, which may correlate with their upregulation in CRC; can promote the metabolic changes in cancer cells (switch to aerobic glycolysis) [37].	Diagnosis and mechanisms
miRNAs	***miR-675***	H19-derived miRNA was upregulated in cell lines and primary CRC as compared to noncancerous tissues through downregulation of RB, increased tumor cells growth, and regulation of the CRC development [176].	Mechanisms and therapeutics
	**miR-483**	miR-483 was a dominant driver oncogene at the *IGF2* 11p15.5 CRC amplicon, inducing dysplasia in vitro and tumorigenicity in vivo [109].	Mechanisms and therapeutics
	**miR-483-3p, miR-483-5p**	The levels of IGF2, miR-483-3p, and -5p were synchronously increased in CRC tissues; IGF2 correlated with both types of miRNAs; and higher smiR-483-5p levels were found compared to controls [143].	Diagnosis, mechanisms, and therapeutics
	**miR-486-5p**	Plasma miR-486-5p expression was upregulated in CRC; decreased levels were associated with TNM stage, larger tumor size, lymphatic metastasis, and poor prognosis [101].	Diagnosis, mechanisms, prognosis, and therapeutics
	**miR-491-5p**	The overexpression of IGF2 rescued the miR-491-5p-induced suppression of proliferation in CRC cells; decreased plasma miR-491-5p expression in CRC was found compared to controls [177].	Mechanisms, prognosis, and therapeutics
	**miR-181a/135a/302c**	DNA methylation mediated repression via repressing PLAG1/IGF2 signaling; promotes the microsatellite-unstable CRC development and 5-FU resistance [178].	Mechanisms, prognosis, and therapeutics

AS: antisense; CRC: colorectal cancer; CRIS: CRC intrinsic subtype; CRNDE: Colorectal neoplasia differentially expressed; CSCs: cancer stem cells; CTCF: CCCTC-Binding Factor; eIF4A3: Eukaryotic initiation factor 4A-III; FSCN1: Fascin Actin-Bundling Protein 1; iDEGs: imprinting-associated Differentially Expressed Genes; lncRNAs: long non-coding RNAs; LOH: loss of heterozygosity; LOI: loss of imprinting; miRNAs: mikroRNAs; MSI: Microsatellite instability; OR: odds ratio; RB: retinoblastoma; shRNA: short hairpin RNA; s: serum; TADs: topologically associating domains; wt: wild type.

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
