# Peer review of "Insulin-Like Growth Factor 2 (IGF2) Signaling in Colorectal Cancer—From Basic Research to Potential Clinical Applications"

_ijms, 2019, doi:10.3390/ijms20194915_

Round 1
Reviewer 1 Report
The paper “Insulin-Like Growth Factor 2 (IGF2) signalling in colorectal cancer – from basic research to clinical practice” summarizes the many information in the literature on the role of growth factor IGF2 in colorectal cancer.
The authors collected a great deal of material and organized it in a fairly coherent way. However, precisely the large amount of information, sometimes contradictory, and the many varied aspects in which IGF2 is implicated, makes the paper quite complicated, sometimes confused and difficult to follow. The addition of further schemes and figures could help the reader to extricate himself from this complex information network and to better focus and understand the role of IGF2 in CRC.
Regarding the general organization, the part on the actual and potential role of IGF2 in the treatment of CRC is less developed than the biology, physiology and the role of this molecule in the pathology of CRC.
Apart from this general comment, there are some critical points that I list below.
Abstract: the sentence “While genetic, as well as epigenetic……..is unknown” sounds contradictory; please better explain and/or reword. The same sentence is present also in the introduction.
Introduction, “Among the other 46 CRC patients…”: I don’t understand the sense of this sentence and the link with the previous ones of the same paragraph in which neither patients nor their numbers were referred to.
Paragraph 2.1, “IGF2, first isolated…, together with the second polypeptide…”: it is not clear to me which is the second polypeptide referred to.
Paragraph 3.1: it is not clear if there is a difference between autocrine and paracrine effects and what they are. The effect of IGF2 on CRC cells should be the same whether secreted by tumour cells or by cells of the microenvironment.
Paragraph 3.2: the title can be misunderstood. In fact, this paragraph deals mainly with the mechanism that induce the overexpression of IGF2 and only marginally, as I would have expected, with the mechanisms of action of IGF2 responsible for the induction of carcinogenesis in CRC and its progression.
The sentence “Some of the data…..in a sex-specific manner” makes no sense as formulated.
Paragraph 3.4. Epidemiological evidences seem to be rather contradictory in different literature studies. Can the authors better comment on that?
Paragraph 3.5. “There is a discussion…….or worse survival prognosis” these couple of sentences seem contradictory.
The part of “clinical practice” is practically absent.
Paragraph 5. “This epigenetic alteration even….”: this sentence has no sense.
Some minor English revision is required here and there.
Author Response
Dear Reviewer,
We wish to thank you very much for a favourable review, all critical remarks and time spent on reviewing the manuscript.
Regarding the potential role of IGF2 in the treatment of CRC – sparsity of this subchapter is caused by the lack of literature data on the topic. Its main aim is to present the potential courses of combined therapy with IGF2 participation.
Minor remarks:
The potentially misleading sentence (Abstract, Introduction) has been amended, as it suggested poor knowledge of particular pathogenetic mechanisms of most of the CRC cases, despite the description of genetic and epigenetic changes in most of the patients. A sentence on the detection of LOH in CRC patients was also amended.
In paragraph 2.1. – IGF1 is the other polypeptide described in tandem with IGF2- this information was added to the manuscript.
In paragraph 3.1. – this subchapter aimed to emphasize the role of IGF2 produced locally by autocrine (A) or paracrine (B) tumours (without bloodstream participation); by autocrine, the authors relate to the signals affecting the cells that synthetized the hormone, with paracrine defined as those affecting the neighbouring cells. A and P signals exert similar effects, mostly increasing cellular proliferation, maintenance of tumorigenic phenotype of tumour cells or increased local tumour regrowth. It was noted that “tumorigenic clones produce IGF2 at levels 80 times higher than IGF1 and that IGF2 is very likely the main effector in the autocrine loop” [137]. On the other hand, “tumour stromal IGF2, via the paracrine IGF1R/insulin receptor axis, activated pro-survival AKT signalling in CRC cell lines” [136].
The concept of “auto-/paracrine loop of cancer cell autostimulation” is used multiple times in various literature sources [94].
In paragraph 3.2 – In the context of IGF2, the main role in colon carcinogenesis is played by overexpression of IGF2 and direct oncogenic function in CRC cells. Hence, this subchapter aimed to reference the mechanisms responsible for the increase in IGF2 gene/protein activity in CRC. These mechanisms were also summarised in Figure 2 on Figure 2. We changed the title of this subchapter.
IGF2 action through activation of various signalling pathways leading to carcinogenesis was described in detail in many reviews. In this manuscript (to maintain reasonable length) these works were cited in this subchapter and other parts of the manuscript [paragraph 2.0, 3.6].
The description of studies from the cited work (in paragraph 3.2; 3.5, 5.0), as well as the title of the manuscript was modified.
Paragraph 3.4. – a commentary was added to this paragraph.
As per suggestion, the manuscript was thoroughly revised, with all language errors corrected. The publication was corrected by a qualified, native speaker, familiar with the manuscript topics.
All changes (and all additions) in the text were marked red.
Sincerely yours,
Aldona Kasprzak
Reviewer 2 Report
The authors extensively discuss the Insulin-Like Growth Factor 2 (IGF2) signaling in colorectal cancer and provide a general aspect of the research findings from basic to the clinical level. The manuscript is interesting since the components of IGF have gained focus in carcinogenesis research. However, certain points need improvement in order for the authors to deliver a readable and understandable manuscript.
The authors should shorten the abstract and emphasize on the general notion that describes the role of IGF2 in colorectal cancer as well as the potential for therapeutic strategies. The Introduction section is too long. The authors should briefly describe the biology of colorectal carcinogenesis and emphasize on the role of IGF2. The authors analyze the role of IGF2 in the Introduction section. They should transfer the paragraphs from “IGF2… to diagnostic-prognostic factor [15,34]” and incorporate them in respective sections of the manuscript. The authors should describe the role of IGF2 in physiology prior to its role in carcinogenesis. The authors should cite and briefly comment in the Concluding Remarks and Future Directions section (section 5) the following relevant to epigenetic alterations of the IGF2 gene in CRC references: i) Gyparaki MT et al. DNA methylation biomarkers as diagnostic and prognostic tools in colorectal cancer. J Mol Med (Berl). 2013 Nov;91:1249-1256. ii) Mäki-Nevala S et al. DNA methylation changes and somatic mutations as tumorigenic events in Lynch syndrome-associated adenomas retaining mismatch repair protein expression. EBioMedicine. 2019 Jan;39:280-291.Author Response
Dear Reviewer,
We wish to thank you very much for a favourable review, all critical remarks and time spent on reviewing the manuscript.
In abstract and Introduction sections we included the most important discoveries connected with IGF2 signalling. We are aware of the lengthiness of the Introduction section. However, this volume is required considering the complexity of CRC tumours, development of new techniques of carcinogenetic mechanism analysis, as well as discoveries in the field of the activity of many growth factor encoding genes (including IGF system), CRC development factors etc.
In Introduction, a „background” of the complete work was outlined, with the main aims of the review related to IGF2 stated. However, our aim was to include deeper analysis and discussion of the study results in the subsequent chapters. The role of IGF2 in physiology is described in paragraph 2.1 (page 4), purposefully preceded by a few introductory sentences on the of the IGF system in carcinogenesis (with a reminder indicating the signalling pathways). The rest of the manuscript concerns IGF2 in connection to colon carcinogenesis and novel approaches to the discovery of the unknowns in development of this cancer, as well as currently poorly described information on potential anti-IGF2 treatments.
As per Reviewer’s suggestion, both of the literature positions were cited in a small commentary at the end of the manuscript.
As per suggestion, the manuscript was thoroughly revised, with all language errors corrected. The publication was corrected by a qualified, native speaker, familiar with the manuscript topics.
All changes (and all additions) in the text were marked red.
Sincerely yours,
Aldona Kasprzak
Reviewer 3 Report
In the study entitled “Insulin-Like Growth Factor 2 (IGF2) signaling in colorectal cancer – from basic redearch to clinical practice” (Manuscript ijms-601175) the Authors reviewed the role of IGF2 in the pathogenesis of colorectal cancer (CRC), including initiation, progression, liver metastasis, prognosis and possible therapeutic approaches.
This topic is of importance since, as the Authors have addressed, no review of the literature specifically deals with IGF2 and CRC. Accondingly, the comprehension of the molecular mechanisms insight the pathogenesis of CRC would help to suggest novel therapeutic approaches (some of them interestingly provided in the current paper). The Figure is comprehensible and helps the reader in the understanding. Similarly, the Table is well-structured. I would suggest to focus on the following minor points:
To improve the overall organization of the structure of the review since the amount of information provided might make the reading heavy. Minor English editing is required (e.g. “redearch” in the title; 3.6 paragraph, etc.) Page 3. IGF2 is close to H19 (doi: 10.5534/wjmh.190070). The Authors might explain the mechanism by which abnormal IGF2 DMR methylation led to altered IGF2 expression. Does hypomethylation cause IGF2 up-regulation? What about H19 DMR methylation? How does it impact on the IGF2 expression? Finally, might the paternally imprinted nature of the IGF2 gene explain the higher CRC risk in the male gender compared to the female one?
Author Response
Dear Reviewer,
We wish to thank you very much for a favorable review, all critical remarks and time spent on reviewing the manuscript.
Regarding the editorial and other remarks:
„Redearch” in the title was most probably an editorial mistake, which was since corrected, with an amendment of the sub-title in Paragraph 3.6; I know that the IGF2 is close to H19, which is mentioned multiple times in the manuscript, e.g. in 2.1.; 6. section; ref. 35,169,170; When it comes to the mechanism of IGF2 overexpression – it is stated on page 3 that „Loss of imprinting (LOI) of IGF2 could lead to IGF2 overexpression, increased cell proliferation and tumor development [11,28,56,62,67,68,69]”.More details on that topic are present in paragraph 3.6. There is still some discussion on the mechanisms responsible for altered IGF2 expression and IGF2 LOI in CRC [69,140,171,172,173,174]. It is generally thought that overexpression of IGF2 in CRC is mostly caused by LOI of the gene. However, LOI is only observed in some of the patients (ok.30-85%), as well as in normal tissue. Cui et al. (2002) showed that previously described mechanism typical for LOI with hypermethylation of a DMR upstream of the H19 gene, allowing activation of the normally silent maternal allele of IGF2, does not apply to CRC. It has been shown that IGF2 and H19 DMRs in CRC do not follow a reciprocal methylation pattern, suggesting LOI of IGF2 in CRC is associated with H19 and IGF2 DMR hypomethylation [172]. According to the studies by Cui et al., IGF2 LOI (and IGF2 overexpression) is caused by H19 and IGF2 DMR hypomethylation [172]. This work is also cited in the manuscript. Additionally, there is some proof of a possible mechanistic link between alcohol intake and CRC risk through IGF2 DMR0 hypomethylation during colorectal carcinogenesis [71] (page 3). Concerning H19, in general, the sequencing data indicate that CG sites at IGF2 and H19 DMRs in tumors underwent hypomethylation, while sites in normal tissues remain hemimethylated [Cheng et al., 140]. However, there are some works (page 3) indicating IGF2 among the five genes with the highest average hypermethylated percentages (50.4%) in CRC group, and hypomethylated in about 50% control tissues [56].In paragraph 3.6. and Table 1 there are some citations and discussion related to H19 DMR.
It is hard to say if paternally imprinted nature of the IGF2 gene explains a higher CRC risk in the male gender compared to the female. Personally, I consider this process in CRC (a highly heterogenous tumour), in the context of IGF2 signaling connected with other imprinted genes (H19), to be more complex. However, there was not enough space left in this manuscript for proper analysis of this topic.
All changes (and all additions) in the text were marked red.
Sincerely yours,
Aldona Kasprzak